# MATCH PREDICTION USING LEARNED HISTORY EMBEDDINGS

## ABSTRACT

Contemporary ranking systems that are based on win/loss history, such as Elo (Elo, 1978) or TrueSkill (Herbrich et al., 2007) represent each player using a scalar estimate of ability (plus variance, in the latter case). While easily interpretable, this approach has a number of shortcomings: (i) latent attributes of a player cannot be represented, and (ii) it cannot seamlessly incorporate contextual information (e.g. home-field advantage). In this work, we propose a simple Transformer-based approach for pairwise competitions that recursively operates on game histories, in addition to modeling players directly. By characterizing each player by its history, rather than an underlying scalar skill estimate, it is able to make accurate predictions even for new players with limited history. When restricted to the same information as several state of the art methods, including Elo and Glicko (Glickman, 1995), our approach significantly outperforms them on predicting the outcome of real-world Chess, Baseball and Ice Hockey games.

## 1 INTRODUCTION

Systems for ranking teams or individuals in pairwise competitions have important uses in a wide variety of applications that include: (online e)-sports matchups, dating apps, rating sports teams, rating chess players and training populations of agents (Balduzzi et al., 2018).

The most widely used techniques use as input the outcomes of previous match-ups, this being universally available and requiring no domain insights or meta-data. For easy interpretability, most of them use a single scalar, corresponding to a latent skill attribute, to characterize a player or team. This single value is then used to predict the outcome of match-ups or games. However, this implies that the multitude of attributes that makeup a player can be represented by a single number, which is a questionable assumption.

This paper introduces a simple Transformer-based approach for predicting the outcome of competitions from historical data alone. Each player/team is represented by an embedding vector, computed entirely from the history of previous competitions. While a scalar skill rating for each player cannot directly be computed, the model can still be used for the same ranking or match-up tasks as existing methods. While the model is naturally able to incorporate contextual information alongside input history, we focus on using historical data alone, restricting our contextual information to home-field advantage. This is the most general scenario and enables direct comparison to many widely used techniques, avoiding the confounding effect of feature engineering.

A key feature of our model is that it is not necessary to make use of player identity in order to achieve strong predictions: each player is represented entirely by their previous games and outcomes. In a recursive fashion, the opponents of these games are also represented by their history. The purely relational summary of player history allows it to quickly produce accurate predictions for new players for whom little history exists, or for teams at the start of a new season. Player identity can be included in the model, which enables the modeling of non-transitive relations, as we show in a simple example of rock-paper-scissors.

## 2 RELATED WORK

**Skill-based methods:** Skill ranking has a long history in professional sports and competitive games (Glickman, 1995; Dehpanah et al., 2020). In chess, the most widely-known skill rating system is Elo (Elo, 1978). This assumes that a player $A$ has a skill rating $\theta_A$ and that its performance $p_A$ is distributed according to $p_A \sim \mathcal{N}(p_A; \theta_A, \sigma^2)$ where $\sigma$ is some fixed variance. Then the probability that $A$ beats $b$ is given by

$$\Pr(p_A > p_B|\theta_A, \theta_B) = \Phi\left(\frac{\theta_A - \theta_B}{\sqrt{2}\sigma}\right)$$

where $\Phi$ is the cdf of a unit Gaussian distribution with mean 0.

Replacing $\Phi$ with a logistic distribution results in a Bradley-Terry model (Bradley and Terry, 1952). Both Elo and Bradley-Terry can be made Bayesian by assuming some distribution over the skills of each player, i.e. each player has a variance $\sigma_A$ in addition to $\theta_A$ so that each player's performance is modeled as a Gaussian distribution with mean $\theta_A$ and variance $\sigma_A$, rather than a fixed $\sigma$. These modifications are made in the Glicko system (Glickman, 1995) and (Weng and Lin, 2011) to Elo and Bradley-Terry, respectively. Both models can be extended to allow for teams of players and more than two person competitions, which is done in TrueSkill (Herbrich et al., 2007) (Herbrich et al., 2008) and Bayesian-Bradley Terry (Weng and Lin, 2011).

Our approach contrasts with the methods above in that each player is represented by an embedding vector rather than a single number (plus variance). In terms of how player history is utilized, Trueskill (Herbrich et al., 2007), is closest to ours but still differs significant ways: (a) it uses player-specific representations, whereas ours in agnostic to the player ID; (b) it combines information using a graphical model, rather than a deep learning model and (c) it is a flat model, whereas our builds the representation by recursively expanding opponent histories.

**Embedding approaches:** (Delalleau et al., 2012) used a neural network architecture to learn latent embeddings for players, aiming to predict player enjoyment in online matchups. (Zhang et al., 2010) learn a factor embedding for each player and take the inner product with a context vector to get a skill rating. (Balakrishnan and Chopra, 2012) use similar methods for user preferences ("do you prefer A or B?"). Our approach differs in that it focuses on the more general no-context scenario. In this setting the data only consists of outcomes (win/loss or points differential) and time features, better revealing differences in modeling performance.

**Non-transitivity:** Recent work by (Balduzzi et al., 2018) proposed a determinant-based framework for modeling non-transitive interactions between RL agents on a leaderboard. Their approach draws on ideas from combinatorial Hodge theory, as laid out in (Jiang et al., 2011). In the same spirit as (Chen and Joachims, 2016a;b) who use a 1-layer MLP, we instead rely on the inherent non-linearity of deep neural networks to capture transitive and non-transitive relations (if player ID is provided). (Chen and Joachims, 2016c) learn two embeddings for each player, a "blade" vector and a "chest" vector and use the difference between the two to model the probability of winning that captures intransitivity.

**Multi-player:** Another group of approaches explicitly address multiplayer competitions (Herbrich et al., 2007; Minka et al., 2018), e.g. multiplayer online battle area games (MOBA). Many approaches are customized to particular game environments and incorporate domain specific information, e.g. Heros of Newerth (Suznjevic et al., 2015) and Ghost Recon Online (Delalleau et al., 2012). Our method cannot naively handle multi-player settings, but there are plausible approaches to incorporating this, as we discuss later.

**Context:** Many recent approaches have shown ways to incorporate contextual information alongside historical outcomes. (Chen and Joachims, 2016a) uses context, such as betting odds, to improve predictions. TrueSkill2 (Minka et al., 2018) refines TrueSkill (Herbrich et al., 2007) to incorporate context. While not the focus of our paper, since our approach is neural network-based it is able to easily learn useful features from context, if needed.

## 3 LEARNED HISTORY EMBEDDINGS

### 3.1 NETWORK GRAPH

The most intuitive explanation of our approach is from a network graph perspective. Consider a graph where each node $n = (A, B, g)$ represents a game between players $A$ and $B$ along with a node attribute, $g$. We consider the simple case where $g = \{r, t\}$ where $r$ is a scalar indicating the point differential between $A$ and $B$ and $t$ is the time of the game. There is a directed edge between two nodes $n_1$ and $n_2$ defined by $n_1 = (A_1, B_1, g_1)$ and $n_2 = (A_2, B_2, g_2)$ if $\{A_1, B_1\} \cap \{A_2, B_2\} \neq \emptyset$[1] and $t_1 > t_2$. Thus edges in this graph connect nodes that share one player, and neighbors of a node $n$ are defined to go backward in time.

The architecture for our approach relies on Transformer (Vaswani et al., 2017) layers that use attention to learn embeddings of players, operating on an increasingly large receptive field in this network graph. The "level 1" version of our method will operate on neighbors that are path length one away from the proposed match-up; the "level 2" version considers all nodes of path length less than two from proposed match-up, and so on. Note that our model acts only on the node features $g = (r, t)$ so it has no knowledge of explicit player ID in the match-up, thus it is acting only on the histories of the players. This allows for greater flexibility and forces the model to learn richer representations compared to prior work.

### 3.2 LEVEL 1 HISTORY

We wish to predict the outcome of a proposed match occurring between players $A$ and $B$, which will result in either a win or loss (draws are discussed later). Our model makes this prediction using the $T$ previous match-ups of each player $Z_A := \{g_1^A, ..., g_T^A\}$ and $Z_B := \{g_1^B, ..., g_T^B\}$ respectively. These histories are considered level 1 since they correspond to the nodes in the network graph that are depth one away from each player. When computing the probability that $A$ beats $B$ we must respect the underlying symmetry that $p(A \text{ beats } B) = 1 - p(B \text{ beats } A)$. To do this we stack the inputs as $Z_1 = \left( \begin{bmatrix} Z^A \\ Z^B \end{bmatrix} \right)$ and $Z_2 = \left( \begin{bmatrix} Z^B \\ Z^A \end{bmatrix} \right)$ and run them through a multi-layer Transformer $\zeta$ such that $o(A, B) := \zeta(Z_1) - \zeta(Z_2)$. The resulting logit $o(A, B)$ has the desired property $o(A, B) + o(B, A) = 0$ and is trained with cross entropy loss.

This level 1 model is only able to learn fairly simple weighted averages over the history, or generic comparisons of the form "team A is doing better than team B in its past $x$ games".

### 3.3 LEVEL 2 HISTORY

The model becomes more powerful once we consider all nodes in the graph that are at most path length two away from the current node. These nodes correspond to "histories of history", where all previous games in the proposed match-up history include the history of each opponent involved.

Consider two players $A$ and $B$ with $T$ most recent match histories $Z_A = \{g_1^A, ..., g_T^A\}$ and $Z_B = \{g_1^B, ..., g_T^B\}$ respectively. We now also use the $i$th opponent's own history, looking at *their T* previous match-ups. These are represented by features $h_j^i$, from the match-up with the $j$th opponent in player $i$'s history, $i$ being an opponent in $A$'s history.

We learn an embedding for each opponent $i$ in player $A$'s history by passing the features $h_j^i, j \in 1..T$ through a multi-layer Transformer and an MLP. This step summarizes the history of nodes at depth 2 into an embedding vector $\hat{h}^i$. The level 1 history of player $A$ is sumarized using the same Transformer + MLP to produce $\hat{g}$. The level 1 and 2 history embeddings are combined to produce a summary embedding $Z^A$ for player $A$. The equivalent summary for player $B$ is computed in the same fashion using the same set of attention + MLP layers. Figure 1 shows this two level hierarchy.

Formally, for a given player, we have the level 1 history $X \in \mathbb{R}^{T \times k}$, each row of which corresponds to $g_i$, $k$ being the number of features per match-up ($k = 2$ for the case $g = r, t$). We also have level 2 history $Y \in \mathbb{R}^{T \times T \times k}$, which is a tensor that holds $h_j^i$. Both $X$ and $Y$ are processed with

---

[1] i.e. $A_1 = A_2$ or $A_1 = B_2$ or $B_1 = A_2$ or $B_1 = B_2$.

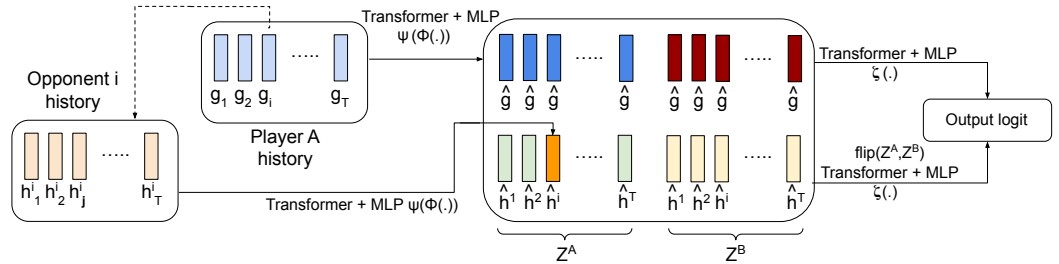

Figure 1: Overview of our two level player history embedding model. See text for details.

the same multi-layer Transformer $\phi(.) : \mathbb{R}^{T \times k} \to \mathbb{R}^{T \times d}$, $d$ being the output embedding dimension (a hyper-parameter). The result is then passed to an MLP which $\psi(.) : \mathbb{R}^{T \times d} \to \mathbb{R}^e$ summarizes the embeddings. Thus, the level 1 and 2 histories are summarized into $e$ dimensional embeddings $\hat{g} = \psi(\phi(X))$ and $\hat{h}^i = \psi(\phi(Y))$, respectively. These are are combined into a matrix $Z$ (of size $2e \times T$):

$$Z = \begin{bmatrix} \hat{g} & \cdots & \hat{g} \\ \hat{h}^1 & \ldots & \hat{h}^T \end{bmatrix}$$

Here the $i$th column corresponds to an embedding representing the $i$th match-up in a player's history. The first part of the embedding, $\hat{g}$, captures the player's own history (level 1), which is shared for each game in the history, and the second part of the embedding, $\hat{h}^i$, represents the history of opponent in the $i$th game.

Computing $Z$ for each player $A$ and $B$ yields $Z^A$ and $Z^B$, which we combine using a multi-layer Transformer $\zeta(.) : \mathbb{R}^{2e \times 2T} \to \mathbb{R}$ to produce a scalar output. Optionally, contextual information $c$ about the encounter between $A$ and $B$ can be added at this stage by appending it to $Z^A$ and $Z^B$. Using the same procedure as in Section 3.2 to enforce the underlying symmetry, output logit $o$ is defined to be:

$$o = \zeta \left( \begin{bmatrix} Z^A \\ Z^B \\ c \end{bmatrix} \right) - \zeta \left( \begin{bmatrix} Z^B \\ Z^A \\ c \end{bmatrix} \right) \tag{1}$$

## 3.4 MODEL ARCHITECTURE DETAILS

In level 2 version of the model, $\phi(.)$ and $\zeta(.)$ each consist of a 3 layer Transformer (see Figure 2). Each of the layers uses the standard architecture from (Vaswani et al., 2017): 4-head attention $\to$ LayerNorm $\to$ 1 layer MLP $\to$ LayerNorm. The output embedding dimensions are $d = 32$ and $e = 8$. Overall, the model is 6 Transformer layers deep ($\phi(.)$ and $\zeta(.)$), plus a single MLP layer ($\psi(.)$).

The level 1 version of the model, which does not look at opponent histories, consists of computing $\psi(\phi(X^A))$ and $\psi(\phi(X^B))$, concatenating them and passing resulting vector through an MLP. $\phi$ and $\psi$ are defined as above.

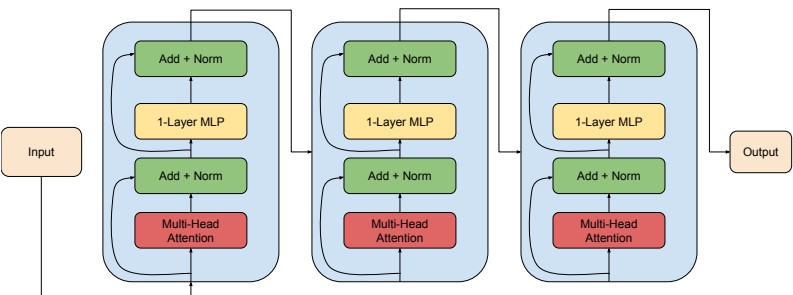

Figure 2: Architecture of the 3-layer Transformer (Vaswani et al., 2017) used for functions $\phi(.)$ and $\zeta(.)$. See text for details.

## 3.5 Practical implementation

It is only necessary to train the model a single time on a fixed data set. The cost of training is linear in the number of matchups (as with Glicko2), being $O(T^2 x M)$ where $M$ is the number of matchups in the data set. Additionally, prediction is fast, being $O(T^2)$ which takes time on the order of milliseconds. Further practical speedups are possible by caching each matchup so that future matchups making use of it can simply look up the associated vector, rather than recomputing afresh. Overall, our approach is still highly performant, albeit slightly slower than Glicko2.

## 4 Experiments

### 4.1 Data sets

Our main evaluation uses four large real-world datasets:

- CHESS1: A data set from the first Kaggle chess competition matches (Kaggle A).
- CHESS2: A data set provided by Kaggle from their from their second competition (Kaggle B).
- HOCKEY: National Hockey League (NHL) data from 1979-2020 (Hockey).
- BASEBALL: Major League Baseball (MLB) data from 1969-1991 (Baseball).

The CHESS1 data set consists of 65053 matches and contains 6573 unique players. Each player in CHESS1 plays an average of 17.8 games in the data set. The CHESS2 data set consists of 366000 games and 7651 unique players. We use a 60000 game subset of this data which has 4529 unique players, each of whom appears an average of 15.7 times. The HOCKEY data set is provided by Hockey Summary Project (Hockey) and spans the years 1979-2020 of regular season and playoff games, during which 41 separate teams play. There are a total of 43984 matches in this data, of which each team appears an average of 2145.5 times. The BASEBALL data set is provided by Baseball Databank (Baseball) and spans the years 1969-1991 of regular season and playoff games. During this period 28 unique teams play an average of 3304 times.

### 4.2 Dataset construction

For CHESS1 we take the first 55000 games to be training, the next 2000 to be validation, and the remaining 8053 used as a test set. For CHESS2 we take the first 50000 games to be training, the next 2000 to be validation, and the remaining 8000 to be the test set. For HOCKEY we take the first 38000 games to be training, the next 2000 to be validation, and the remainder to be the test set. For the BASEBALL we take the first 38000 games to be training, the next 2000 to be validation, and the remaining to be the test set.

The evaluation metric we report in Figure 3 is the Pearson correlation between the prediction and the outcome (0=loss, 1 = win, 0.5 = draw), which for two variables $X, Y$ is defined as

$$\rho_{X,Y} = \frac{\text{cov}(X,Y)}{\sigma_X \sigma_Y}.$$

This allows us to easily compare with the continuous variable EMA baseline (described below), which does not naturally form a probability. In chess, if there is a draw, we take the target value $y$ to be 0.5 in the log-loss used for training:

$$p \log y + (1 - p) \log(1 - y),$$

effectively asking the probabilities to be close to 0.5.

### 4.3 Experiment configuration

The hyper-parameters were tuned using the validation set and are described in Table 1.

| | Environment | | | |
|---|---|---|---|---|
| | CHESS 1 | CHESS 2 | BASEBALL | HOCKEY |
| Learning rate | .0005 | .0005 | .00025 | .00025 |
| Optimizer | Adam | Adam | Adam | Adam |
| Number epochs | 10 | 10 | 6 | 16 |
| Batch size | 120 | 120 | 40 | 40 |
| T | 40 | 40 | 40 | 100 |
| Embedding size $d$ | 32 | 32 | 32 | 32 |
| Embedding size $e$ | 8 | 8 | 8 | 8 |
| EMA half-life | 500 | 500 | 120 | 180 |

Table 1: The hyper-parameters for our model in each of the environments

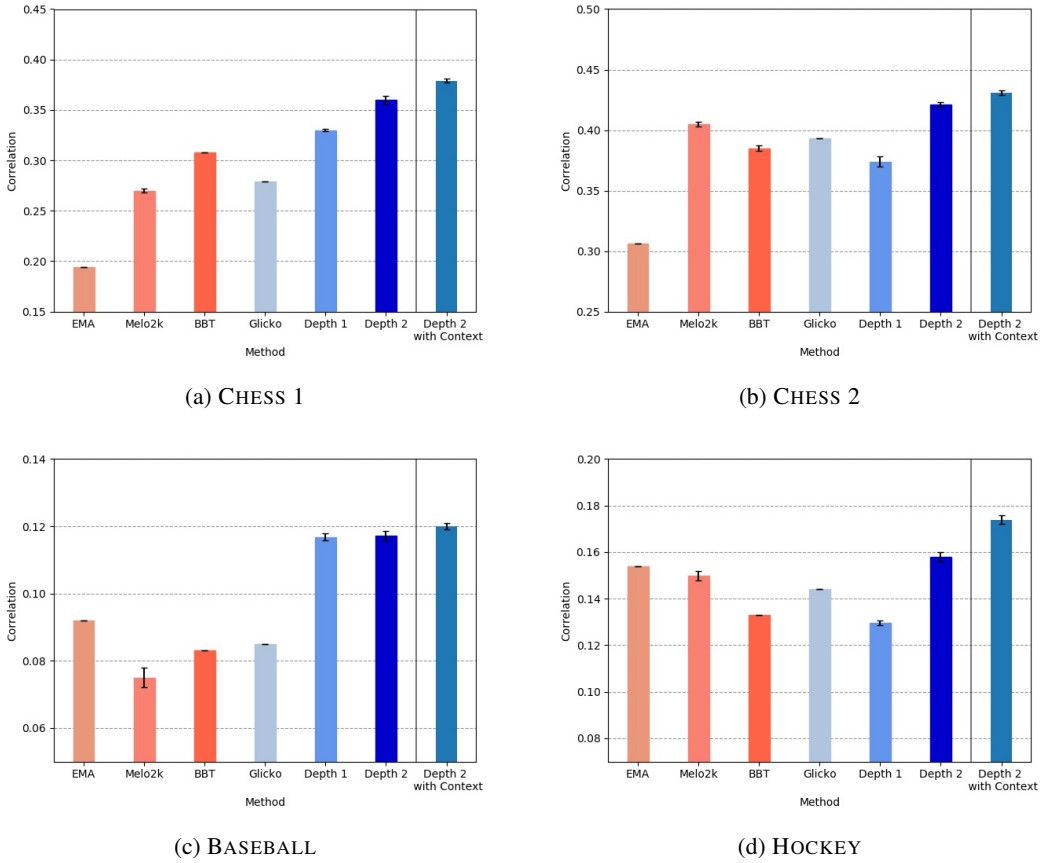

(a) CHESS 1

(b) CHESS 2

(c) BASEBALL

(d) HOCKEY

Figure 3: Match prediction on the evaluation set of four real-world datasets: CHESS1, CHESS2, BASE-BALL, and HOCKEY. Pearson correlation with the true outcome of each game is plotted. Our learned history embeddings approach is shown in blue. Depth 1 corresponds to the architecture described Section 3.2, and depth 2 corresponds to the architecture described in Section 3.3. Exponential Moving Average (EMA), Bayesian Bradley-Terry (Weng and Lin, 2011), Blade-Chest (Chen and Joachims, 2016c), Melo2K (Balduzzi et al., 2018), and Glicko (Glickman, 1995) baselines are shown in orange. Error bars show $\pm 1\sigma$ for five seeds with different model initializations. Our method significantly outperforms the baselines, with depth 2 models generally being superior to depth 1. On the right side of each subplot we show a depth 2 model with context added (black/white for CHESS1/CHESS2; EMA of point differential for BASEBALL and HOCKEY). These features boost the model performance over the context-free version.

### 4.4 BASELINES

We compare our approach to the following baselines: (i) Glicko2 (Glickman, 2013), (ii) Bayesian Bradley-Terry (Weng and Lin, 2011), (iii) an exponential moving average (EMA) of the win loss (in chess) and the point differential (in baseball and hockey). EMA is a statistic on a sequence of values $(V_1, ..., V_T)$ that keeps track of a recency weighted mean. The EMA $m_t^\alpha$ at time step $t$ depends on $\{V_i\}_{i \leq t}$ and is defined as

$$m_t^\alpha = \alpha m_{t-1}^\alpha + (1 - \alpha)V_t$$

where $\alpha$ is a hyper-parameter. The hyper-parameters used for Glicko2 and BBT are listed in the appendix. We use the open-source R package "Sequential Pairwise Online Rating Techniques"(Kałedkowski, 2020). The hyper-parameters for each baseline are tuned on the validation set. We use the author's publicly available implementation of the Blade-Chest model (Chen and Joachims, 2016c), but we find it does not perform well on the four data sets, achieving correlation of numbers 0.010, 0.077, 0.012, 0.023 on CHESS1, CHESS2, BASEBALL, and HOCKEY respectively.

### 4.5 RESULTS

The results are shown in Figure 3. Each approach is evaluated using 5 different random seeds for model initialization and show $\pm 1\sigma$ error bars. In all four environments the depth 2 method performs the best out of the models which only look at win/loss and point differential data, significantly outperforming Glicko2 and BBT baselines.

When additional meta-data is added to the depth 2 model, performance is boosted for all four environments. In CHESS1/CHESS2, the context consists of telling the model which player is playing white or black. In BASEBALL and HOCKEY, this context consists of adding the EMA prediction as context for each player.

For hockey and baseball data, the exponential moving average of point differential is a strong baseline, and it outperforms both Glicko and BBT. The strength of EMA in the sports setting can likely be explained by the fact that at any point in time each team has played roughly the same number of games, so fast adaptation to recent results is less informative than in chess. Despite the strength of these baselines we still handily beat them with our approach, showing the good inductive bias of treating the history of games as a graph.

Each dataset we examine is temporal in nature (i.e. the match ups follow a distinct ordering in time), so there is only one possible choice for the training/test split. In other words, doing cross validation with random splits would lead to cheating since having knowledge of future games can leak information into what happened in the past. In particular, in chess simply knowing that a player plays in the future leaks information about its past performance, since there is a bias for winners to continue playing and losers to drop out. This requires evaluating our model using a moving block bootstrap method on the sequential data. In the baseball domain, over five folds, sliding the training window up a season at a time (162 games), our method outperforms Glicko2 (Glickman, 2013) by an average of 0.021 with a standard deviation of 0.006 and outperforms BBT (Bradley and Terry, 1952) by an average of 0.029 with a standard deviation of 0.009.

### 4.6 FAST START EVALUATION

In Figure 4 we compare our model to BBT for differing numbers of previous games. We look at the minimum number of games played by each player and look at the correlation between the prediction and the response for different buckets of this value. For both CHESS1 and CHESS2, the model is strong relative to the BBT when it has to quickly adapt to new players who have only played a few games. This contrasts with traditional methods such as BBT, Glicko, and EMA that require a burn-in period in order to adjust to the player.

### 4.7 NON-TRANSITIVITY

While the experiments above make no use of player ID, explicitly adding learning a representation per player enables the modeling of non-transitive matchups. We demonstrate this by modeling the game of rock-paper-scissors using the above model enhanced with a player ID feature. Trained on a toy rock-paper-scissors data set, our model correctly predicts the outcome of the various matchups. By

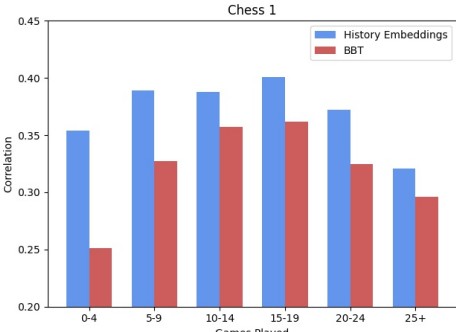 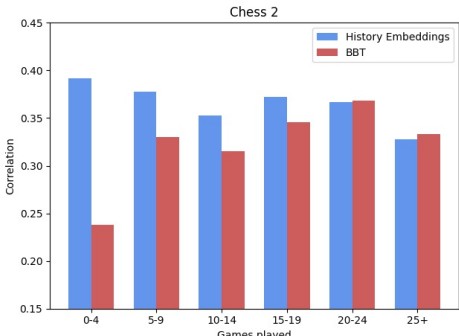

Figure 4: A breakdown of our performance as a function of history size, i.e. number of match-ups for each player in the evaluation set (min over each pairing) on the CHESS1 (L) and CHESS2 (R) datasets. We see that our approach (blue) is able to make accurate predictions for new players even with very few games ($< 10$), where as the performance of Bayesian Bradley Terry (BBT, red) degrades in such settings.

contrast, Glicko (Glickman, 2013) and the scalar based methods are no better than random. Melo2K (Balduzzi et al., 2018) and BladeChest (Chen and Joachims, 2016c) are capable of modeling this intransitive behavior as well.

In the four real-world data sets, we find that adding player ID as a feature does not significantly improve the performance of our model, suggesting that the non-transitive effect is small.

## 5 DISCUSSION

We have introduced a novel and conceptually simple model for learning from pairwise comparison data. This framework and architecture are able to capture complex inter-dependencies between players' histories and learn robust representations that are useful in predicting the winner of matches. The experiments show significant gains against widely-used skill rating approaches that use game history information alone.

The model also has several qualities that make it practical for real-world applications: First, the fast start ability, shown in Figure 4, is vital for many online ranking systems, where new players must be reliably matched to appropriate opponents, otherwise they will quickly lose interest in the game. Second, at inference time our approach is very fast since the model is lightweight, using a modest number of feed-forward layers. Furthermore, it is based on standard deep learning architectures for which many software and hardware optimizations exist. This makes it feasible to run the model on environments with very large numbers of players.

Code and trained models are available at `https://anonymous.4open.science/r/ltc_cam-C52A/README.md`.

### 5.1 LIMITATIONS

The model, as described, does not compute a scalar skill quantity for each player, making it difficult to interpret. However, it is possible to estimate an approximate measure by randomly sampling other players and averaging the computed win probability against each. This would result in a scalar in the range 0 (lose to everyone) to 1 (win against everyone).

### 5.2 FUTURE WORK

The simplicity of the framework allows for enhancements in a number of directions.

- Improved architectures: A natural direction is to recurse deeper into the player interaction graph by using level $n$ versions of our model, which would grow the receptive field. Extending our model to 3 levels is straightforward, as the same machinery can be used to compute

embeddings of histories one level deeper. These can then by combined by $\zeta(.)$, with each column of $Z$ being composed of three embedding vectors, rather than the current two.

- Multi-way competitions: While our approach does not immediately generalize to multi-way competitions, there are plausible approaches to doing so. The central issue is that in an $n$-player game, $\binom{n}{2}$ pairwise ordering constraints must be respected. This could be performed by a graph neural-net (which is invariant to input ordering) acting on the $n$ players of the game, each of whom has an embedding produced by our current model.

- Context: Our results in Figure 3 are a proof-of-concept that adding context can improve our models performance. However, the information used (white/black for chess; EMA for ice hockey and baseball) is very basic. If the goal is to produce the best absolute performance for a given sport, then the priority is to source and utilize as much relevant meta-data as possible. For ice hockey and baseball, numerous statistics about each game are typically available, some of which would have useful predictive value. Our model is well suited for learning from such heterogeneous data, given the well-established versatility of Transformer models.

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
