# OpenReview forum: "Match Prediction Using Learned History Embeddings"
_ICLR.cc/2022/Conference — ICLR 2022 Submitted_

### Official Review · Reviewer_eXjB · 2021-10-25

**Correctness:** 3
**Technical Novelty And Significance:** 4
**Empirical Novelty And Significance:** 2
**Recommendation:** 6
**Confidence:** 4

**Details Of Ethics Concerns:**

No concerns.

**Main Review:**

# Strengths

1) Novel, simple approach
	- Sequence modeling has not been used in the past (to my knowledge) for skill rating problems to any success. This is a strong new result even if the components (transformers) are not particularly novel.
	- The approach is also very simple and easy for practitioners to adopt. The relatively efficient prediction computation makes this an attractive option for use in real-world systems.
2) Reproducible results
	- The methods provide sufficient detail to understand the system and the code is provided.
3) Strong initial results
	- (Assuming the results hold under statistical tests, as described below) The results are promising for a method with little additional tuning to the domain task.
	- A strong simple approach is very promising for further refinement and development to address different application domains or problem features. Handling intransitivity is also an important feature that captures a key advantage of other recent skill rating techniques.


# Weaknesses

1) Lack of statistical tests of differences from results
	- Results include standard deviations but no statistical tests for differences. This applies to the plots in Figure 3 and the moving block bootstrap results reported in the text.
	- Without statistical tests readers cannot assess whether results are significant nor the effect sizes of differences.
	- Action: Perform statistical tests for differences among the methods and report significance of differences and estimated effect sizes.
2) Unusual performance metric
	- Pearson correlation is not a standard metric for evaluating predictive systems. While this may be helpful to compare with the EMA baseline the results seem less central than the prediction accuracy. Additionally, in many practical matchmaking systems it is important that predictions be calibrated.
	- The paper would be much stronger with results reported in terms of predictive accuracy, and ideally also including reports of the model calibration curves (ex: If the model predicts a 20% win probability, what fraction of those matches are wins vs losses? If calibrated, 20% of the predicted matches will be wins.)
	- Action: Report results on test set prediction accuracy for all models aside EMA.
		- I would recommend reporting these results as the main outcomes in the paper and moving the correlation results to an appendix for readers interested in the EMA baseline.
	- Action: Report calibration curves for the predictive models (all but EMA)
3) Missing baseline comparisons
	- Fast start evaluations only compare to BBT. These results should include comparisons to the other pairwise comparison methods for fair comparison.
	- Action: Report results for Melo2k, Glicko2, and Blade-Chest on the fast start task.
	- Action: Add statistical testing for differences among the methods.


Note: The results in Figure 3 have a few inconsistencies that make them hard to compare and read. I would recommend the following changes:
1) Use a single range for the y-axis, rather than different ranges for each plot.
2) Include the results from Blade-Chest in the plot. The caption notes these are included but the text simply describes the numbers while omitting them (perhaps to avoid distorting the scale).

Note 2: No results were recorded for the toy domain comparison. Add some simple numeric results or include these in an appendix. The text alone is not sufficient to back the claim made.

**Summary Of The Paper:**

The paper presents an approach to predicting match outcomes between two individuals (teams or players) by comparing embeddings of previous match outcomes of both individuals. Embeddings are generated using a transformer architecture and can be applied recursively to embed the history of opponents of previous matches. Logits are produced by stacking embeddings and training with cross entropy after pushing outputs through an MLP. Experiments compare the method to established pairwise rating algorithms and the results show strong performance on chess, hockey, and baseball datasets. A toy demonstration shows the model can capture intransitive relationships in rock-paper-scissors.

**Summary Of The Review:**

The paper technique is very novel and has clear practical applications. As a practitioner in this domain I would happily use this technique. The key weakness that reduces my score is the lack of rigor in evaluating the results. This makes it difficult to tell how strong the results are.

I would move my score up if the authors provide statistical evidence of improvements, particularly if using prediction accuracy, rather than correlation coefficients.

---

> ### Author Response · Authors · 2021-11-29
> **Response**
>
> Thank you for the positive review. We agree that a more comprehensive set of evaluation metrics would be helpful and will add them to future versions of the paper.

---

### Official Review · Reviewer_ciZW · 2021-10-31

**Correctness:** 3
**Technical Novelty And Significance:** 3
**Empirical Novelty And Significance:** 3
**Recommendation:** 3
**Confidence:** 4

**Main Review:**

Strengths:
Different from Glicko which only maintains a scalar representing the player’s skill, this paper utilizes player’s embedding for match prediction. This deep learning model can also enable the flexibility to combine context information into the prediction process.

Weaknesses (questions):
1- The appendix is cited but not attached to the submission, which contains hyperparameter settings.

2- Given the constructed logit o in Eq.(1), how do you generate the probability P(A beats B)? It is easy to guess that the Bradly-terry model is employed, but I would appreciate it if the authors could explicitly define it as there are temperature parameters to be specified.

3- From the methodology of constructing the embeddings in Sec. 3, I do not see how the proposed method can be ensured to deal with intransitivity.

For example, if A= Rock, B= Scissors, what value the logit o will be in Eq.(1); is that zero?

If so, the predicted P(Rock beats Scissors ) will be 0.5 instead of 1. Could you explain the details of dealing with intransitivity?

4- In Sec 3.2, the cross entropy loss is not given. Though it is known to many readers, I would appreciate it if the authors could expand the details instead of relying on the readers’ preliminary knowledge on this.

5- While Glicko can be used to predict the match outcome of cold start players, can this method be generalized to cold start players too?

6- A related question to Question 2.
Sec 4.7 this paper claims the capability of modeling non-transitive games, but no empirical evidence is provided.


**Summary Of The Paper:**

This paper aims at predicting the winning probability of a pairwise comparison. It adopts a deep neural network to combine historical comparison results and generate embeddings of two competitive players to make the final prediction. For each player, this paper considers two-levels of historical information preserved in a graph network. Experiments show that compared with traditional Glicko2 algorithms, the deep learning method has higher predictive accuracy.

**Summary Of The Review:**

This paper is easy to follow and has practical merits. However, it lacks the formality of an academic paper, with jumbled symbols, important elements that are not explicitly defined, sections and paragraphs that are too short, etc. I would recommend the authors to make careful revision and correction.

---

> ### Author Response · Authors · 2021-11-29
> **Response**
>
> 1. We apologize for not including the appendix. We only tune one hyper-parameter in the baselines BBT and Glicko, and that is the initial standard deviation of player rating (referred to as init_rd in the sport package [18]). For the EMA baseline the only parameter is the half-life.
>
> |      | Glicko rd | BBT rd     | EMA half-life |
> | :---        |    :----:   |          ---: | ---: |
> | Chess1      | 350       | 20 this   | 1000   |
> | Chess2   | 350        | 20      | 1000      |
> | Baseball   | 29.16        | 1      | 160      |
> | Hockey   | 50        | 8.33     | 160      |
>
>
> 2. To turn the logit into a probability we use the softmax function. We will clarify this.
> 3. (& 6.) We respectfully feel the reviewer may have misunderstood our claims. We agree that our model cannot model intransitivity if player IDs are absent from the histories. Indeed, we make this clear in both the Introduction and Section XX. However, in Section 3.7 we explore a different scenario to previous experiments, namely what happens if we now *do incorporate* player ID into the histories. Using the toy rock-paper-scissors game, we show the model is able to perfectly model this scenario, one which clearly requires the modeling of intransitivity. We will include performance numbers to make this point more clearly.
> 4. We will include a definition of cross-entropy loss.
> 5. We are not sure exactly what is being asked: Glicko ratings are well-defined at any point in a player's career, so it can already deal with cold-starting. It could be possible to increase the performance of Glicko on limited data, but that is out of the scope of this paper.

---

### Official Review · Reviewer_TCF7 · 2021-11-01

**Correctness:** 3
**Technical Novelty And Significance:** 2
**Empirical Novelty And Significance:** 2
**Recommendation:** 3
**Confidence:** 4

**Main Review:**

**Pros**

- The main advantage of the proposed algorithm is the representation of each player with an embedding vector that is learned based on its playing history. Actually, no further information is used that reveals player identity.
- Two different versions have been proposed. The first one learns only an embedding vector for each player according to her history. On the other side, the second one also learns a separate embedding for each opponent in player's history (level 2). Then the level 1 and level 2 embedding are combined to represent the player.
- Experiments have been conducted on four real-world datasets

**Cons and Comments**

- Some parts of the paper are not well written, making it hard for the reader to understand the proposed algorithmic scheme. Specifically, Section 3 should be revised carefully. For instance, in Section 3 the connection between Network Graph and the proposed algorithm is not clear. Also, $g^A$ seems to refer to the final outcome of a game but $g=(r,t)$ on the previous section refers to the difference on the score ($r$) between the two players at time $t$.
- The setting of the experiments is not clear. Actually, the way where the dataset is split into training/validation/testing sets is possible to reveal information about the agents. It would be more reasonable to split the dataset based on the players. Another way would be the training of the algorithm on the Chess 1 dataset and its testing on the Chess 2.
- The evaluation metric that is used is only the Pearson correlation. Why is the log-loss not used for the evaluation of the performance of the proposed algorithm?  Minor comment: formula of the cross entropy at Section 4.2 is not correct, please fix it.
- It seems that the performance of the proposed algorithm is decreased as the history size is increased. Authors should discuss in details why does it happen.
- Melo2k is one of the baselines that have been used,  but it is not presented at Section 4.4.

**Summary Of The Paper:**

This paper introduces a Transformer based algorithm for predicting the outcome of a match or a game. The main novelty of this work is the representation of each player with an embedding vector that is learned based on the player's history (previous played games and their outcomes). Then, based on the learned embedding of the two players that participate in the game, the proposed algorithm predicts the output of the game. Experiments have been conducted on four real-world datasets.

**Summary Of The Review:**

This work presents some interesting ideas but it is not ready for publication in its current form. As aforementioned, the empirical analysis is not complete and the authors should revise the manuscript carefully. For all these reasons, I recommend rejecting the paper.

---

> ### Author Response · Authors · 2021-11-29
> **Response**
>
> 1. We disagree with the claim that "the dataset is split into training/validation/testing sets is possible to reveal information about the agents". The temporal split we use prevents future information being used by the agent, i.e. the model during training is only given information before a certain cutoff date and validation/testing utilizes information after that date. This enforces a disjunction between the train & test sets. By contrast, the dataset split proposed by the reviewer *would* in fact leak future information about players between the train/test, thus cannot be used.
> 2. Thank you for pointing out this inconsistency in notation. We will address it and endeavor to make the algorithmic section more clear.
> 3. We use Pearson correlation rather than log-loss so that we can directly compare to the EMA baseline, which is just a scalar. We will revise the paper to include EMA run through a logistic regression so that we can compare log-loss for all of the methods.
> 4. We agree that more evaluation metrics would be helpful and will add them in a future iteration of the paper.

---

### Decision · Program_Chairs · 2022-01-20

**Decision:**

Reject

**Comment:**

This paper presents a new method that uses transformers to predict the result of pairwise competitions given each players’ history of past game plays. The reviewers thought this had notable potential benefits for practice. However the reviewers’ also had some significant concerns with the current work in terms of the evaluations used, which were primarily  correlation instead of prediction accuracy or calibration etc. There was also some concern about other aspects of the presentation. We hope that the reviewers’ responses are useful to the authors in revising their work for future submissions as this method has the potential to be very useful for many domains.